# Epicortical Brevetoxin Treatment Promotes Neural Repair and Functional Recovery after Ischemic Stroke

**DOI:** 10.3390/md18070374

**Published:** 2020-07-21

**Authors:** Erica Sequeira, Marsha L. Pierce, Dina Akasheh, Stacey Sellers, William H. Gerwick, Daniel G. Baden, Thomas F. Murray

**Affiliations:** 1Department of Pharmacology and Neuroscience, Creighton University, Omaha, NE 68123, USA; EricaSequeira@creighton.edu (E.S.); mpierc1@midwestern.edu (M.L.P.); dinaakasheh@creighton.edu (D.A.); staceysigmon@gmail.com (S.S.); 2Center for Marine Biotechnology & Biomedicine, Scripps Institution of Oceanography, San Diego, La Jolla, CA 92093, USA; wgerwick@ucsd.edu; 3Center for Marine Science University of North Carolina Wilmington, Wilmington, NC 28409, USA; baden@uncw.edu

**Keywords:** brevetoxin, ischemic stroke, peri-infarct, neuroplasticity

## Abstract

Emerging literature suggests that after a stroke, the peri-infarct region exhibits dynamic changes in excitability. In rodent stroke models, treatments that enhance excitability in the peri-infarct cerebral cortex promote motor recovery. This increase in cortical excitability and plasticity is opposed by increases in tonic GABAergic inhibition in the peri-infarct zone beginning three days after a stroke in a mouse model. Maintenance of a favorable excitatory–inhibitory balance promoting cerebrocortical excitability could potentially improve recovery. Brevetoxin-2 (PbTx-2) is a voltage-gated sodium channel (VGSC) gating modifier that increases intracellular sodium ([Na^+^]i), upregulates N-methyl-D-aspartate receptor (NMDAR) channel activity and engages downstream calcium (Ca^2+^) signaling pathways. In immature cerebrocortical neurons, PbTx-2 promoted neuronal structural plasticity by increasing neurite outgrowth, dendritogenesis and synaptogenesis. We hypothesized that PbTx-2 may promote excitability and structural remodeling in the peri-infarct region, leading to improved functional outcomes following a stroke. We tested this hypothesis using epicortical application of PbTx-2 after a photothrombotic stroke in mice. We show that PbTx-2 enhanced the dendritic arborization and synapse density of cortical layer V pyramidal neurons in the peri-infarct cortex. PbTx-2 also produced a robust improvement of motor recovery. These results suggest a novel pharmacologic approach to mimic activity-dependent recovery from stroke.

## 1. Introduction 

Ischemic stroke is a common neurological disorder and major cause of long-term disability worldwide [1]. Currently, tissue plasminogen activator (tPA) is the only FDA-approved pharmacologic treatment for ischemic or thrombotic stroke, which carries the risk of producing an intracerebral hemorrhage [2,3,4]. Although this pharmacologic advancement of acute care has resulted in a decline in mortality rate, it has produced a greater number of disabled survivors. Soon after stroke onset, oxygen-deprived neurons in the infarct core cease to function while tissue in the surrounding peri-infarct region remain viable but compromised [5]. Previous work has suggested parallels between plasticity mechanisms in the developing brain and those occurring in the adult brain after a stroke event [6,7,8,9,10]. Neuronal circuits do undergo limited re-mapping and reorganization after stroke, and these repair processes are associated with neurogenesis, dentritogenesis, synaptogenesis, axonal sprouting and rewiring of cortical networks in the peri-infarct tissue [11]. However, this spontaneous reorganization only partially restores motor function. To more fully regain recovery of motor function, additional pharmacologic manipulations in the peri-infarct are required. 

Glutamate-mediated excitotoxicity has been shown to contribute to ischemic cell death due to failure of ionic homeostasis and a sustained elevation of intracellular calcium concentration [12]. Following the excitotoxicity-induced acute insult, there is a period of recovery with characteristic heightened neuroplasticity in the peri-infarct tissue [13]. It is therefore critical that pharmacologic treatments to promote recovery be administered subsequent to the acute phase of the stroke. In rodent stoke models, pharmacologic and genetic strategies that enhance neuronal excitability in the peri-infarct cortex adjacent to the stroke promote motor recovery [14]. These mechanisms that enhance neuronal plasticity are similar to those involved in learning and memory [13]. In this regard, it is noteworthy that N-methyl-D-aspartate ionotropic glutamate receptors (NMDARs) are crucial in activity-dependent synaptic changes and in learning and memory.

Previous research has shown that changes in intracellular sodium concentration ([Na^+^]_i_) produced in the soma and dendrites as a result of neuronal activity may act as a signaling molecule and play a role in activity-dependent synaptic plasticity. Synaptic stimulation elevates [Na^+^]_i_ to 10 mm in dendrites and up to 35–40 mm in dendritic spines [15]. In hippocampal neurons, such intracellular [Na^+^] increments have been demonstrated to increase NMDAR-mediated whole-cell currents and NMDAR single-channel activity by increasing both channel open probability and mean open time [16].

Brevetoxins (PbTx-1 to PbTx-10) are potent lipid soluble polyether neurotoxins produced by the marine dinoflagellate *Karenia brevis* [17]. PbTx-2 interacts with neurotoxin site 5 on the α subunit of voltage-gated sodium channels (VGSCs), and augments sodium influx by inhibiting channel inactivation and shifting the activation potential to more negative values [18]. Src kinases are widely expressed in the brain and regulate activities of ion channels such as NMDARs. Phosphorylation of NMDAR tyrosine residues by Src facilitates the binding of Na^+^ to NMDAR and exerts a regulatory effect on NMDAR signaling [19]. Single-channel currents recorded from cell-attached patches on cerebrocortical neurons indicate that PbTx-2 upregulates NMDAR whole-cell currents by increasing mean open time and probability without affecting the resting membrane potential [20]. This upregulation is attributed to the coincident elevation of intracellular [Na^+^] and Src kinase activation [21]. PbTx-2 treatment of cerebrocortical neuron cultures robustly potentiated NMDAR-mediated calcium influx (Ca^2+^) [20]. In immature cerebrocortical neurons, PbTx-2 treatment enhanced neurite outgrowth, dendritic arborization, synaptogenesis and filopodia formation and maturation [22]. In addition, PbTx-2 exposure engaged downstream activity-dependent mechanisms involved in neuronal growth and survival such as Ca^2+^-calmodulin kinases (CaMKs), extracellular signal-regulated kinase (ERK), cAMP response element binding protein (CREB) and brain-derived neurotrophic factor (BDNF) signaling pathways [22]. PbTx-2 exhibited a characteristic bidirectional concentration–response profile similar to that of NMDA since an optimal window for [Ca^2+^]_i_ is required for neurite extension and branching [23]. Inasmuch as the mechanisms involved in repair processes after stroke are similar to those regulating neuronal development, we hypothesized that PbTx-2 may augment recovery following ischemic stroke. 

We therefore explored neurohistochemical and functional outcomes of administration of PbTx-2 during the recovery phase after stroke. To assess neurohistochemical changes, we imaged neurons in the peri-infarct cortex to assess dendritic arborization and synaptic density. In humans, long-term disabilities related to stroke often include impairments in feeding, coordination and gait. To examine impairment and recovery, we utilized a catwalk test to examine gross motor gait, a pasta matrix reach task to assess fine-motor skills (feeding and coordination) and a foot fault task to examine coordination and gait. An emerging strategy in stroke therapy is the direct application of treatments to the stroke lesion [24,25,26]. Accordingly, we mixed PbTx-2 in a hydrogel composed of thiol-modified hyaluronan and polyethylene glycol diacrylate and this composite was deposited epicortically directly above the stroke cavity. We demonstrate that epicortical application of PbTx-2 at five-days post-infarct enhances neuronal repair and improves functional outcomes.

## 2. Results

### 2.1. PbTx-2 Enhances Neuronal Structural Plasticity in the Peri-Infarct Region as Revealed by Increased Dendritic Arbor Complexity and Synapse Formation

Using 2- to 4-month-old male yellow fluorescent protein (YFP) line-H transgenic mice, we produced unilateral photothrombotic strokes by providing an intraperitoneal injection of a light sensitive dye followed by exposure of the motor cortex region to a cold source of light. Mice were then allowed to recover in home cages for five days (Figure 1). On day 5, animals were treated with the vehicle or PbTx-2 and subsequently sacrificed on day 6 to examine neuronal structural changes in the peri-infarct region. 

We performed cresyl violet staining to evaluate the extent of the photothrombotic stroke volume and the effect of epicortical application of PbTx-2 on the same (Figure 2A). Coronal sections of 100 μm were collected using a vibratome (Leica VT 1200S). For the infarct volume measurement, brain sections were stained with 0.5% cresyl violet and images were then acquired using a bright-field microscope. The areas of infarct were delineated and quantified using the Image J software (NIH) and infarct volume was calculated by summation of the lesion areas of all sections and integrated by the thickness of the section (Figure 2B). Infarct volumes did not vary significantly between vehicle- and PbTx-2-treated mice (Figure 2C). These results indicate that the timing of PbTx-2 treatment utilized was appropriate for the assessment of promoting functional recovery inasmuch as these treatments did not affect stroke volume.

Dendritic injury is a pathologic hallmark of excitotoxicity inasmuch as NMDARs are predominantly localized on dendrites [27,28]. Stroke caused the deterioration of neurons in the infarct core that could be readily distinguished from the adjacent surviving peri-infarct region. Confocal images of layer V YFP-labelled pyramidal neurons in the peri-infarct region were obtained and 3D morphological analysis of the arbor complexity with a defined algorithm was performed using the Imaris image analysis software (Figure 3A). In vehicle-treated animals, there was a gradual increase in branching complexity moving away from the soma, reaching a maximum of 8 ± 0.5 intersections per neuron, and then progressively declining beginning at approximately 10 µm from the soma (Figure 3B). Doses of 10, 100 and 1000 pmols of PbTx-2 produced a 2-fold increase in the expansion of dendritic arbors of neurons in the peri-infarct cortex, and a rightward shift in the Sholl plot as compared with the vehicle-treated control mice (Figure 3B). An AUC analysis of Sholl data showed a significant increase in dendritic complexity following the 10, 100 and 1000 pmol doses of PbTx-2 compared with vehicle controls. The PbTx-2 effect on dendritic complexity displayed a bidirectional profile as shown by a lack of effect on the expansion of dendritic arbors at the 3 and 3000 pmol doses (one-way ANOVA followed by Dunnett’s post hoc test, **** *p* < 0.0001; 3 pmol PbTx-2 effect, *p* = 0.579; 3000 pmol PbTx-2 effect, *p* = 0.998; n = 17 to 51 neurons; Figure 3C). Given our previous demonstration of bidirectional PBTx-2 concentration–response profiles in cerebrocortical neurons similar to that of NMDA, a possible explanation for PbTx-2’s bidirectional profile herein is the underlying NMDAR-dependent mechanism of action [23]. Previous studies have established an inverted-U concentration–response profile for the relationship between NMDAR and neuronal survival, where too little or too large activation of NMDARs can respectively diminish neuronal growth or cause cell death [29].

Next, we examined the influence of PbTx-2 on synapse formation in the peri-infarct cortex region. Confocal images of YFP-labelled neurites in the peri-infarct region were obtained and analyzed using a spot detection algorithm again using the Imaris image analysis software (Figure 4A). Antibodies against VGLUT1 (presynaptic marker) and PSD-95 (postsynaptic marker) were used to quantify synapse density, as revealed by colocalized fluorescent puncta (Figure 4B). We obtained the ratio of synaptic puncta on the ipsilesional side to that of the contralesional side to correct for between animal variation. Dose–response analysis of the effect of PbTx-2 on synapse density indicated that 3, 10, 100 and 1000 and 3000 pmol PbTx-2 produced a significant increase in synapse formation (number of puncta per length of neurite) compared with the vehicle-treated animals (one-way ANOVA followed by Dunnett’s post hoc test, **** *p* < 0.0001; n= 12 to18 neurons; Figure 4C).

Further, to better visualize the PbTx-2 effect size, we compared mean difference (MD) distribution between the vehicle- and PbTx-2-treated groups using the Gardner–Altman mean comparison plot that affords transparency of the effect size of treatments [30] (3 pmol PbTx-2 MD = 37.6%, 95% CI [28.3, 53.4]; 10 pmol PbTx-2: MD = 63%, 95% CI [47.4, 86.2]; 100 pmol PbTx-2: MD = 52.9%, 95% CI [42.3, 60.7]; 1000pmol PbTx-2: MD = 51%, 95% CI [40.3, 60.6]; 3000 pmol PbTx-2: MD = 38.7%, 95% CI [32.4, 47.0]; Figure 4D,E). The effect sizes and CIs are reported as: effect size [CI width—lower bound, upper bound] and the narrowness of the confidence interval represents the effect size precision.

### 2.2. PbTx-2 Promotes Recovery of Fine Motor Skills in Stroke Affected Mice

We next assessed the influence of the photothrombotic focal stroke on motor function and coordination by first performing a CatWalk gait analysis. We examined contralateral forelimb function post-stroke as motor disability in response to photothrombotic stroke [31]. We analyzed several gait parameters on days 1, 2 and 5 post-stroke and compared the stroke ipsilateral left front (LF) paw to the contralateral right front (RF) paw in each animal. No differences between the sham-operated group and photothrombotic stroke (PTS) group were detected. Shown are the results for parameters “max contact”, “intensity” and “stride length” (unpaired two-tailed test: max contact (%), *p* = 0.721; intensity (%), *p* = 0.987; stride length (%), *p* = 0.471; n = 9 mice; Figure 5). These results demonstrated a lack of effect of the stroke on gait parameters and that motor deficits might be confined to the digits of the paw. We therefore next used a pasta matrix reach task and a foot fault test to assess fine motor skills.

Rodents live in an environment that requires the use of a complex range of motor skills to gain access to food [32]. The pasta matrix reach task is one of the few motor tests that can measure skilled forepaw use [33]. Mice were trained for three days to grab a single piece of pasta to determine paw preference (Figure 6A), followed by seven days of pasta matrix training sessions to obtain baseline values before inducing stroke (Figure 6B). On day 1, following the stroke, there was a significant decline in the number of pasta pieces retrieved, indicating that the photothrombotic stroke affected fine motor skills. On day 6, following the stroke, animals treated with the 10, 100 and 1000 pmol doses of PbTx-2 exhibited a significant increase in the number of pasta pieces retrieved as compared with the vehicle-treated mice. The lowest and highest doses of PbTx-2, 3 and 3000 pmol, were however without a significant effect (one-way ANOVA followed by Dunnett’s post hoc test, **** *p* < 0.0001, *** *p* = 0.0002; 3 pmol PbTx-2 effect, *p* = 0.955; 3000 pmol PbTx-2 effect, *p* = 0.873; n = 9 to 15 mice; Figure 6).

To further confirm the influence of PbTx-2 on functional recovery, we used a foot fault task that represents a sensitive method for detecting motor deficits of limb functioning and placement during locomotion. Animals without a stroke should place their paws precisely on the wire frame and demonstrate few to zero foot faults [33]. The photothrombotic stroke in the forelimb motor cortex produced a significant increase in the percentage foot faults on day 1 after the insult, indicating a disruption of limb function and placement (Figure 7A). At PbTx-2 doses of 10, 100 and 1000 pmols, treated animals displayed significant improvements in percentage of foot faults as compared with the vehicle-treated mice. Again, the 3 and 3000 pmol doses of PbTx-2 did not promote functional recovery (one-way ANOVA followed by Dunnett’s post hoc test, 10 pmol PbTx-2 effect; * *p* = 0.023, 100 pmol PbTx-2 effect; * *p* = 0.027, 1000 pmol PbTx-2 effect; ** *p* = 0.001; 3 pmol PbTx-2 effect, *p* = 0.703; 3000 pmol PbTx-2 effect, *p* > 0.999; n = 9 to 13 mice; Figure 7B). These data establish that treatment with the 10, 100 and 1000 pmol doses of PbTx-2 on day 5 post-stroke resulted in functional recovery of fine motor skills in both the pasta matrix handling and foot fault tasks as compared with the vehicle control treatments.

## 3. Discussion

Here, we investigated the effect of PbTx-2 on neuroplasticity in the peri-infarct cortex and associated motor functions in a murine model of stroke. The main findings of this study are: 1. epicortical application of PbTx-2 at the stroke site produced a 2-fold increase in dendritic arborization and increased synaptogenesis in the peri-infarct cortex, 2. photothrombotic stroke in the forelimb motor cortex produced functional deficits that were confined to the digits of the paw, 3. PbTx-2 doses of 10, 100 and 1000 pmols produced dramatic improvements in functional recovery toward pre-stroke controls as measured by an increase in the number of pasta pieces retrieved or decreased percentage of foot faults and 4. PbTx-2 displayed bidirectional dose–response profiles where the 3 and 3000 pmol doses did not affect neurite outgrowth or motor functional recovery, consistent with these effects being mediated through NMDARs.

VGSCs play a fundamental role in electrical signaling of the nervous system and action potential generation [34]. Two-photon imaging studies show that synaptic stimulation leads to transient increases in [Na^+^]_i_ in postsynaptic spines and dendrites [15]. This suggests that [Na^+^]_i_ may function as a signaling molecule and play a role in activity-dependent synaptic plasticity. PbTx-2, a VGSC gating modifier, augments NMDA receptor signaling through coincidence of an elevation of [Na^+^]_i_ and Src kinase activity [21]. A previous report suggested that PbTx-2-mediated activation of sodium channels was associated with enhancement of NMDA-induced Ca^2+^ influx, accelerated spine formation and maturation, increased dendritic arbor elaboration and increased synaptogenesis in developing cerebrocortical neurons [22]. The cell signaling mechanisms underlying these responses involved PbTx-2-induced increase in intracellular Ca^2+^ with attendant phosphorylation of Ca^2+^-dependent molecules including CaMKI, CaMKII and CREB that play essential roles in neuronal growth and survival. BDNF is an activity-dependent neurotrophic factor that mediates neuroplasticity and is regulated by CREB-dependent mechanisms [35], and PbTx-2 exposure also increased the surface expression of BDNF-tropomyosin-related kinase B receptors in cerebrocortical neurons [22].

Glutamate plays an essential role in mediating excitatory neurotransmission in the central nervous system and is vital for synaptic plasticity. After an ischemic stroke, however, glutamate accumulation leads to excitotoxicity due to over-activation of NMDARs and neuronal death [36,37]. Interestingly, NMDAR antagonists failed clinically to show neuroprotective effects and, in some cases, worsened stroke outcomes in patients [38,39,40,41]. Hence, blocking NMDARs subsequent to a stroke is detrimental inasmuch as glutamate signaling through NMDARs contributes to neuronal survival. This influence of glutamate on NMDARs to promote neuronal survival displays an inverted U-shaped concentration–response curve, where too little or excessive activation of NMDARs are detrimental [42].

Neuronal excitability after a stroke exhibits distinct phases during stroke progression and recovery [43]. In the acute phase, excessive glutamatergic activity produces excitotoxicity and is deleterious. During the subsequent chronic phase however, glutamatergic excitability in the peri-infarct cortex is correlated with neuronal repair and recovery [43]. Therefore, enhancing cortical excitability too early after stroke may further increase neuronal death. This inflection point from the acute excitotoxic to chronic recovery phase occurs three days post-stroke in mice [14]. In the present study, we therefore selected the time point of five days after stroke for the epicortical treatments. Stroke recovery has been associated with dramatic spine plasticity in the peri-infarct cortex and with an increase in dendritic spine density over baseline values in some regions [11]. This influence of glutamatergic signaling is opposed by a marked increase in extracellular γ-aminobutyric acid (GABA) levels due to the loss of GABA transporter GAT-3 [14]. Notably, administration of L655,708, a benzodiazepine inverse agonist specific for extrasynaptic GABA_A_ receptors, produced a rapid and sustained improvement in functional recovery in mice following a photothrombotic stroke [14]. Hence, counteracting the hypo-excitability caused by heightened GABAergic inhibition could potentially promote recovery when initiated during the chronic phase. The present results using the sodium channel gating modifier brevetoxin may provide an additional approach for enhancing brain excitability during the period of recovery and reorganization to promote neural repair. 

We selected the photothrombotic stroke model because it produces a localized infarct that permits a detailed analysis of neuronal structural plasticity and functional recovery [44]. The effect of photothrombotic stroke on forelimb fine motor deficits, as assessed by the pasta matrix reach and foot fault tasks, appeared to be both pervasive and persistent since at six days post-stroke, the vehicle-treated animals displayed profound deficits in task performance. We found that a single epicortical PbTx-2 treatment applied five days post-stroke was sufficient to promote functional recovery and that these beneficial effects were paralleled by PbTx-2-induced increases in dendritic arbor complexity and synaptic density in the peri-infarct cortex. These actions of PbTx-2 on neuronal plasticity and functional recovery both showed inverted-U dose–response curves. We have shown previously that the in vitro effect of PbTx-2 on neurite outgrowth in cerebrocortical neurons exhibited a bidirectional concentration–response (inverted-U) profile and that this effect was primarily dependent on NMDARs [20]. Similarly, the effects of PbTx-2 on dendritic arborization and synaptogenesis in cerebrocortical neurons displayed bidirectional concentration–response profiles [22]. The inverted-U model for the relationship between NMDAR activity and neuronal survival and growth is well established [29]. The inverted-U dose–response effects of PbTx-2 on neuronal plasticity in the peri-infarct cortex observed in the present study are consonant with those of a previous report that neuronal activity affects structural plasticity in vivo through NMDAR-triggered intracellular signaling events [45]. It is therefore reasonable to posit that the effects of epicortical PbTx-2 on structural plasticity and post-stroke functional recovery are the result of elevated [Na^+^]_i_ and enhanced NMDAR function.

Our results demonstrate impairment in forelimb fine motor control in mice after a photothrombotic stroke and reversal of these deficits by PbTx-2 treatment. These data suggest that stroke-induced motor deficits might be particularly responsive to augmented cortical excitability during the recovery phase of stroke. Currently, the only clinical treatment following a stroke is tissue plasminogen activator (tPA) which must be administered within the first few hours post-stroke. Considering that occupational and physical therapy are the standard of care for stroke recovery, our results suggest that sodium channel gating modifiers may represent a novel pharmacotherapy to accelerate recovery. This new strategy to enhance cortical excitability during the delayed time frame important for neural repair and recovery may hold promise for reducing the severity of stroke disability. 

## 4. Materials and Methods

### 4.1. Animals

All animal use protocols were approved by the Institutional Animal Care and Use Committee of Creighton University and employed measures to minimize pain and discomfort. Hemizygous Cg-Tg (B6.Cg-Tg(Thy1-YFP)HJrs/J (yellow fluorescent protein (YFP)-expressing mice) were obtained from Jackson Laboratories (stock 3782) and maintained as a breeding colony. Mice were genotyped by a commercial vendor (Transnetyx, Cordova, TN). Murine *thy1*-YFP line-H strain was selected since it displays bright fluorescence in layer V pyramidal neuron of the motor cortex, providing high contrast, thus facilitating imaging and analysis of fine neuronal structures [46]. Six-days post-infarct under isoflurane anesthesia, (3–4% in a 70% NO_2_ / 30% O_2_ mixture) mice were transcardially perfused with phosphate-buffered saline (PBS), followed by 4% paraformaldehyde (PFA) solution and tissue was isolated for analysis and post-fixed overnight with PFA at 4 °C. 

### 4.2. Photothrombotic Model of Focal Stroke

Photothrombosis is less invasive and produces highly localized and reproducible lesions in comparison with other ischemic stroke models [44]. Adult *thy1*-YFP mice (Jackson Laboratories) aged 2–4 months old were anesthetized with isoflurane inhalation (Piramal Healthcare; VetEquip) and placed in a stereotaxic apparatus (Stoelting). Body temperature was maintained at 36.9 ± 0.4 °C using a TP700 water warming system (Stryker). A midline incision was performed, the skull exposed, and connective tissue removed. Mice were weighed and 100 mg/kg dose of Rose Bengal solution (10 mg/mL in sterile PBS) was injected intraperitoneally (IP). Five minutes after IP injection, the cortex was illuminated 1.5 mm lateral from bregma through an intact skull for 12.5 minutes using a light source (150 mW, KL 1500 LCD, Zeiss) attached to an UPlanApo20x objective (Olympus). Rose Bengal produces reactive oxygen intermediates under light excitation, leading to platelet adhesion and aggregation, which occludes vascular endothelium and results in formation of an infarct [47]. The scalp was closed using Vetbond adhesive (3M, St. Paul, MN, USA) and animals were treated for five days with oral sulfamethoxazole and trimethoprim (SMZ/TMP; 200/40 mg).

### 4.3. In Vivo Epicortical Drug Administration 

Five-days post-infarct, adult thy1-YFP mice (Jackson Laboratories, Bar Harbor, ME, USA) aged 2–4 months old were anesthetized with isofluorane inhalation (Kent Scientific; Vetflow, Torrington, CT, USA) and placed in a stereotaxic apparatus (Stoelting, Wood Dale, IL, USA). Body temperature was maintained at 36.9 ± 0.4 °C using a TP700 water warming system (Stryker, Kalamazoo, MI, USA). The midline incision was reopened, and a small burr hole drilled (Dremel) through the skull 1.5 mm lateral to bregma. PbTx-2 was dissolved in dimethyl sulfoxide (DMSO) (Sigma-Aldrich, St Louis, MO, USA) and then diluted in 0.1X PBS. Vehicle controls were prepared using DMSO in 0.1X PBS (0.1% v/v final concentration of DMSO). The injectate consisted of Hystem-HP (Advanced BioMatrix, Carlsbad, CA, USA) hydrogel mixture (Glycosil + Gelin-s +Extralink) with PbTx-2 or vehicle alone. Vehicle, 3, 10, 100, 1000 or 3000 pmols of PbTx-2 were injected using a Hamilton syringe 0.75 mm below the surface of the brain in a volume of 3 μL at a rate of 0.3 microliters/minute for a period of 10 min. The scalp was closed using Vetbond adhesive (3M, St. Paul, MN, USA) and animals were treated for 1 day with SMZ/TMP. 

### 4.4. Assessment of Dendritic Arbor Complexity

Here, 100 μm brain sections were collected using a vibratome (Leica VT 1200) and mounted onto slides using Gelvatol mounting medium. Ten neurons per animal were used for quantification. We captured 40X images of YFP-labelled layer V pyramidal neurons in the peri-infarct region using a TCS SP2 confocal microscope in z-stack images (Leica Microsystems, Wetzlar, Germany). Each neuron selected for analysis met the following two criteria: (1) neuron was located in the peri-infarct region and (2) neuron was distinct from surrounding neurons for measurement of dendritic arborization. Following image acquisition, the computer-based cell tracing software Imaris image analysis (Bitplane, Zurich, Switzerland) was used for 3D reconstruction of neurons. For the analysis of dendritic arbor complexity, dendritic tracings were quantified by the Sholl analysis method, where a set of concentric spheres (1 μm apart) centered from the cell body were drawn, and the number of dendritic branch intersections at each sphere was quantified and plotted. 

### 4.5. Synapse Density Quantification 

Free-floating slices of 20 μm were washed three times in 0.1 M PBS and blocked in PBS buffer containing 10% goat serum in 0.25% Triton X-100 (Sigma-Aldrich) for 1 h. For synaptic staining, primary antibody solution was added consisting of PSD-95 antibody (1:300, Invitrogen) and guinea pig polyclonal anti-VGLUT1 antibody (1:1000, EMD Millipore). Slices were incubated in primary antibodies overnight at 4 °C, washed three times in PBS and 0.25% Triton X-100, and incubated with Alexa Fluor 594 goat anti-rabbit (1:500, Invitrogen, Carlsbad, CA, USA) and DyLight 405-conjugated AffiniPure Goat anti-guinea pig (1:500, Jackson ImmunoResearch). After incubation in secondary antibodies for 2 h, sections were washed and mounted onto slides using Fluoromount mounting medium (Sigma-Aldrich). 

Images of immunostained brain sections were collected on a Leica TCS SP2 confocal microscope using a 63× oil immersion objective in z-stack images. Serial optical sections at 0.15 μm intervals over a total depth of 10 μm for a total of 3 sections per brain were imaged. Channels for YFP, vesicular glutamate transporter 1 (VGLUT1) and postsynaptic density 95 (PSD-95) were acquired sequentially to prevent spectral overlap of fluorophores. Gain thresholds and amplitude offsets were kept constant between imaging of the ipsilateral and contralateral cortices. A ratio of synaptic puncta on the ipsilateral hemisphere to the contralateral hemisphere was obtained to minimize between-animal variation. Further, a spot detection algorithm in the Imaris image analysis software was used to quantify synapse density (analysis was normalized to number of synapses per neurite segment). 

### 4.6. CatWalk Gait Analysis

Data acquisition: This test was conducted in the dark as mice were required to walk on a glass plate, which was illuminated from below by a fluorescent light. Light enters at the long edge of the plate and is completely internally reflected and escapes only at those areas where the animal’s paw makes a contact. A video camera, positioned underneath the glass plate, captures the illuminated areas and sends the video image to a computer running the CatWalk software (version 7.1, Noldus). All mice underwent training one week prior to receiving a stroke to habituate to the testing environment and procedure. Three runs per mouse were recorded before stroke and on days 1, 2 and 5 post-stroke and the average was used for analysis. 

Footprint classification: After acquiring the video images of paw prints, labels were assigned to illuminated areas (right front (RF), right hind (RH), left front (LF), left hind (LH)). Following footprint classification, the CatWalk software automatically calculates a variety of parameters. A series of gait statistics was produced automatically when the system identified and marked each footprint made. We then compared the ipsilateral LF paw to the contralateral RF paw in each animal of sham-operated (IP injection of PBS) and photothrombotic stroke (PTS)-affected mice (IP injection of Rose Bengal). 

### 4.7. Pasta Matrix Reach Task

To facilitate training, animals were food-restricted for two weeks so that their body weight gradually reached 85% of that compared with ad libitum access to food. Animals were placed in a plexiglass cylinder and trained for 30 minutes to reach through a 1 cm wide slot to grasp, break and retrieve pieces of vertically oriented pasta (Angel Hair Pasta, American Beauty) inserted into a 1mm diameter grid-ordered field of holes. To initially encourage reaching and to determine paw preference, animals were trained to grab a single piece of pasta for 3 days. Once the preferred paw was determined, the matrix was offset such that animals could retrieve pasta pieces only with their preferred paw. Additionally, the matrix was arranged such that it consisted of a 5 × 5 vertical alignment of 5 cm-long segments of uncooked pasta. Subsequent training sessions for a period of 7 days led to improved success in the task and established a baseline. Photothrombotic stroke was induced in the brain hemisphere contralateral to the preferred paw. Performance was quantified by manual inspection of the matrix at the end of the session (counting the number of broken pasta pieces). 

### 4.8. Foot Fault Test

Each mouse was placed individually on the elevated wire grid and allowed to freely walk for a period of 5 minutes. Video footage was analyzed and the total number of foot faults for each limb along with the total number of non-foot-fault steps were counted, and a ratio between foot faults and total steps taken calculated. Percent of foot faults was calculated as follows: number of foot faults / (foot faults + number of non-foot-fault steps) × 100. A step was considered a foot fault if it was not providing support and when the foot went through the grid hole [14]. To minimize variability, a ratio between foot faults and total steps was determined to account for differences in the degree of locomotion between animals and trials. 

### 4.9. Data Analysis 

All data were presented as mean values ± SEM and plotted using GraphPad Prism® 7.0 (GraphPad, La Jolla, CA, USA). Area under the curve (AUC) of Sholl plots was used to assess the number of dendritic intersections for the vehicle and PbTx-2 treatment groups. A Gardner–Altman plot was generated using the Estimation statistics software to compare mean differences between vehicle-treated and PbTx-2-treated groups [30]. The differences in gait assessed by CatWalk between sham-operated mice and stroke-affected mice were examined using an unpaired t (two-tailed) test. Percent of foot faults was calculated as the number of foot faults/(foot faults + number of non-foot-fault steps) × 100. A one-way ANOVA followed by Dunnett’s post hoc comparison test was used to compare 3 or more groups. The results were considered statistically significant if *p* < 0.05 according to the ANOVA. The *p* values, statistical analyses used, and sample sizes are described in the figure legends. 

## Figures and Tables

**Figure 1 marinedrugs-18-00374-f001:**
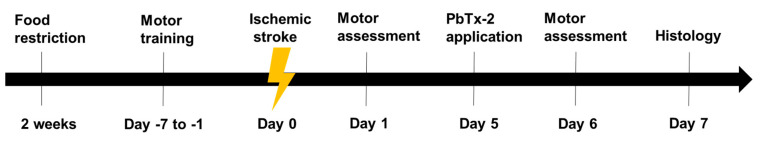
Experimental timeline: Animals were food-restricted to 85% of their body weight for two weeks prior to training for pasta matrix reach task. Animals were trained to perform pasta matrix reach task and foot fault task prior to inducing stroke to obtain baseline values. A focal lesion was induced in the motor cortex region by photothrombosis on day 0. Animals were randomly divided into PbTx-2 or vehicle treatment groups. On day 5, PbTx-2 mice were treated with 3, 10, 100, 1000 or 3000 pmol PbTx-2, whereas vehicle-treated animals were given hydrogel alone. Further, on day 1 (post-stroke) and day 6 (post-treatment), animals were assessed for the number of pasta pieces retrieved and percentage of foot faults. On day 6, animals were sacrificed, and brains were isolated for histological analysis of surviving neurons in the peri-infarct region.

**Figure 2 marinedrugs-18-00374-f002:**
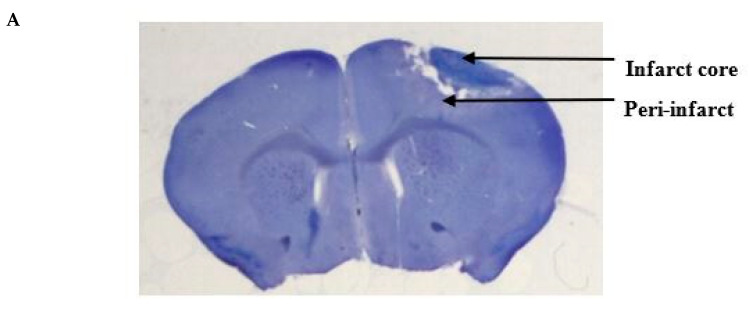
Histologic assessments after stroke. (**A**) Representative cresyl violet image showing the infarct and the peri-infarct region. (**B**) Representative cresyl violet stained-sections of vehicle, 3, 10, 100, 1000 and 3000 pmol PbTx-2, respectively. (**C**) Quantification of infarct volume. No significant differences in infarct size were detected (one-way ANOVA followed by Dunnett’s post hoc test, 3 pmol PbTx-2 effect, *p* = 0.993; 10 pmol PbTx-2 effect, *p* = 0.907; 100 pmol PbTx-2 effect, *p* > 0.999; 1000 pmol PbTx-2 effect, *p* = 0.873; 3000 pmol PbTx-2 effect, *p* = 0.975). Data shown are mean ± SEM of 2 to 7 brains.

**Figure 3 marinedrugs-18-00374-f003:**
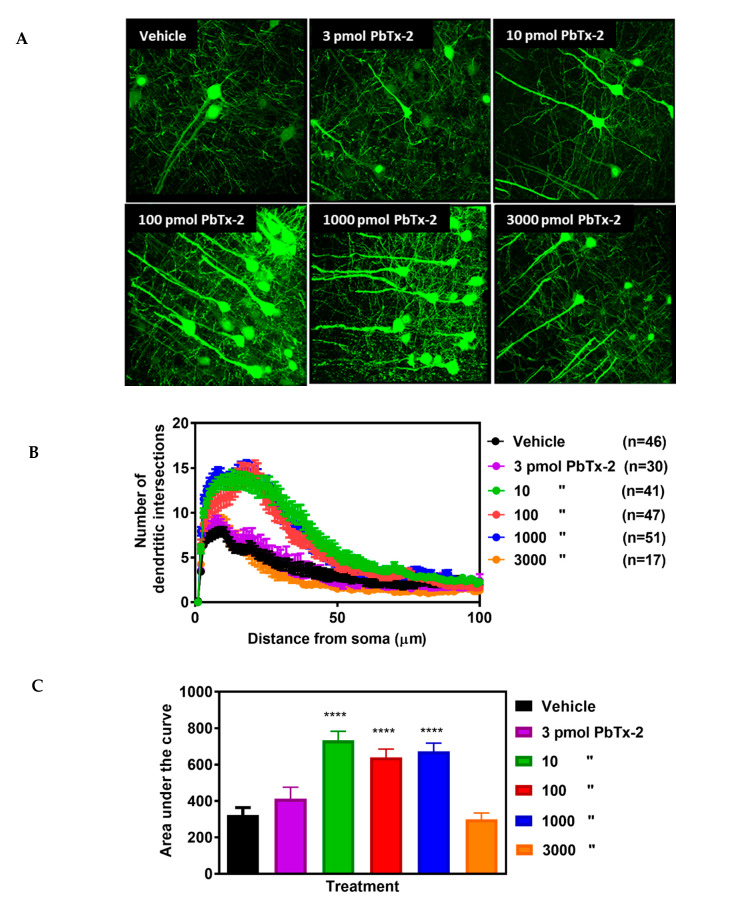
Effect of PbTx-2 on dendritic arborization in the peri-infarct region. (**A**) Representative images of PbTx-2-induced dendritic arborization in the peri-infarct region (Scale bar: 30 μm). (**B**) Sholl analysis to quantify dendritic arbor complexity. (**C**) Area under the curve (AUC) analysis of Sholl data, PbTx-2 at 10, 100 and 1000 pmol doses enhanced dendritic arbor complexity in the peri-infarct site as compared with the vehicle-treated animals (one-way ANOVA followed by Dunnett’s post hoc test, **** *p* < 0.0001). However, the 3 and 3000 pmol doses of PbTx-2-treated mice were without effect on the expansion of dendritic arbors (3 pmol PbTx-2 effect, *p* = 0.579; 3000 pmol PbTx-2 effect, *p* = 0.998). Each bar represents the mean ± SEM of 17–51 neurons.

**Figure 4 marinedrugs-18-00374-f004:**
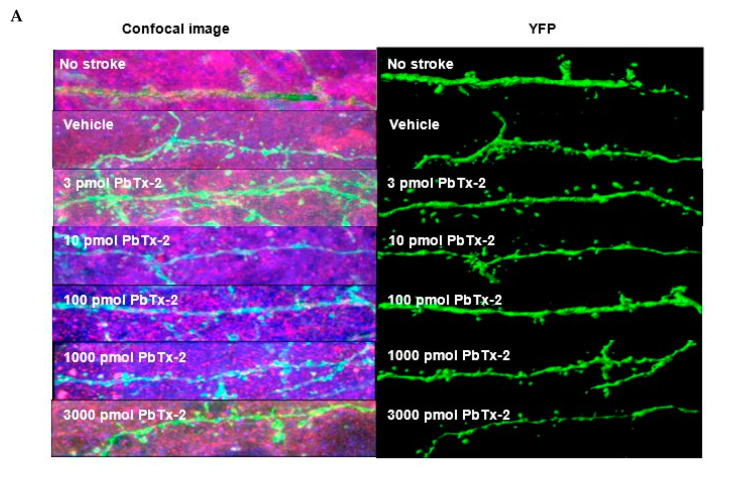
Effect of PbTx-2 on excitatory synapse density in peri-infarct region. (**A**) Representative images of double-immunostained YFP expressing neurites in the peri-infarct region at day 6 obtained from a confocal microscope. (**B**) Antibodies against VGLUT1 (presynaptic marker/blue) and PSD-95/red (postsynaptic marker) were used to quantify synapse density, as indicated by colocalized fluorescent puncta (yellow), scale bar: 5 µm. (**C**) Quantification of colocalized fluorescent puncta using Imaris image analysis; 3, 10, 100, 1000 and 3000 pmol PbTx-2 doses enhanced synapse density in the peri-infarct region (one-way ANOVA followed by Dunnett’s post hoc test, **** *p* < 0.0001). (**D**) Gardner–Altman mean comparison plots showing PbTx-2-induced increments in synapse density. Each plot contains comparisons for all PbTx-2-treated groups, and each dot represents % ipsilateral/contralateral puncta from an individual brain section. (**E**) Gardner–Altman plot demonstrating effect size. The left ordinate axis of the plot shows mean difference (MD) distribution between the PbTx-2-treated and vehicle-treated groups; 3 pmol PbTx-2 MD = 37.6%, 95% CI [28.3, 53.4]; 10 pmol PbTx-2: MD = 63%, 95% CI [47.4, 86.2]; 100 pmol PbTx-2: MD = 52.9%, 95% CI [42.3, 60.7]; 1000pmol PbTx-2: MD = 51%, 95% CI [40.3, 60.6]; 3000 pmol PbTx-2: MD = 38.7%, 95% CI [32.4, 47.0 All data are represented as mean ± SEM of 12–18 brain sections.

**Figure 5 marinedrugs-18-00374-f005:**
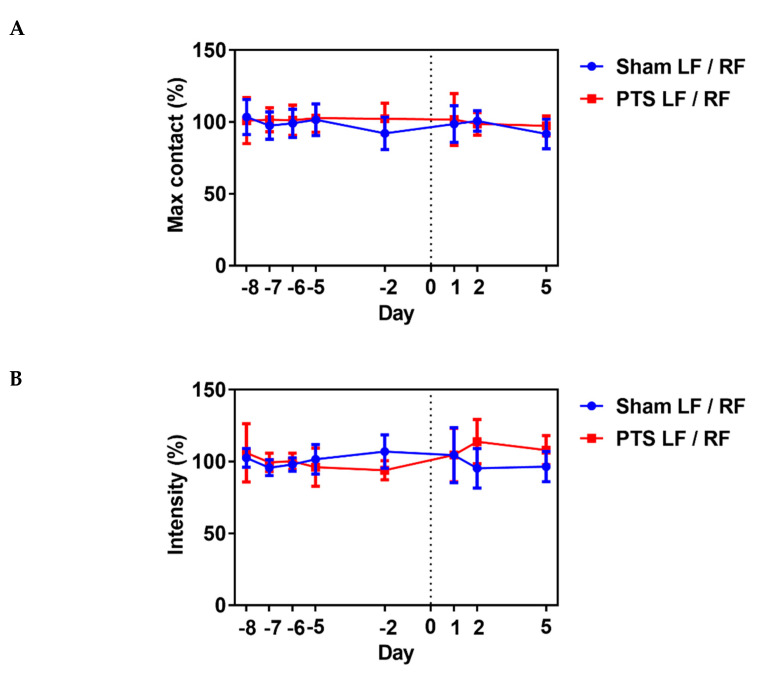
Impact of photothrombotic stroke on gross motor function. Representative CatWalk parameters (**A**) max contact (%), (**B**) intensity (%) and (**C**) stride length (%) analyzed before inducing photothrombotic stroke (PTS) to obtain baseline and on days 1, 2 and 5 after stroke. We compared the ipsilateral left front (LF) paw to the contralateral right front (RF) paw. Gait parameters were not altered by photothrombotic stroke. Shown are the results of unpaired two-tailed test (max contact (%), *p* = 0.721; intensity (%), *p* = 0.987; stride length (%), *p* = 0.471). All values are given as mean ± SEM (n = 9 mice).

**Figure 6 marinedrugs-18-00374-f006:**
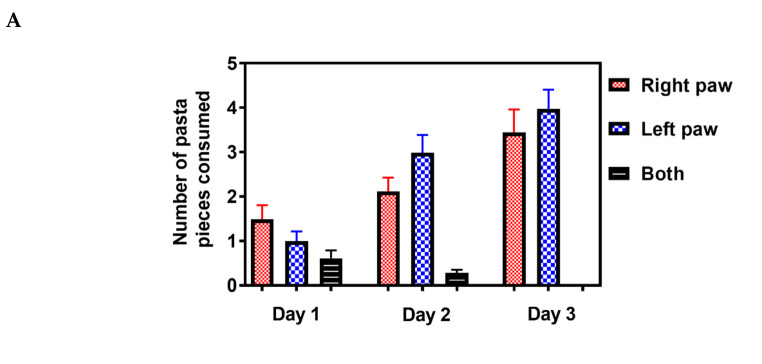
Effect of PbTx-2 treatment on pasta matrix reach task. (**A**) Paw preference was determined using a single piece of pasta for 3 days. (**B**) Animals were trained for 7 days to retrieve 25 pieces placed in a 5X5 matrix from their preferred paw. Day 0 represents the photothrombotic procedure and at 1 day after the stroke, there was a significant reduction in the number of pasta pieces retrieved, indicating motor deficits induced by the insult. On day 6, after the stroke, the 10, 100 and 1000pmol PbTx-2-treated animals exhibited significant increases in the number of pasta pieces retrieved as compared with the vehicle-treated animals. The 3 and 3000 pmol PbTx-2 doses did not facilitate functional recovery. (**C**) Quantification of PbTx-2 dose–response effects on motor recovery 6 days post-stroke (one-way ANOVA followed by Dunnett’s post hoc test, **** *p* < 0.0001, ****p* = 0.0002; 3 pmol PbTx-2 effect, *p* = 0.955; 3000 pmol PbTx-2 effect, *p* = 0.873). All data are represented as mean ± SEM (n = 9 to 15 mice). Data points without error bars are due to the error bar being smaller than the symbol.

**Figure 7 marinedrugs-18-00374-f007:**
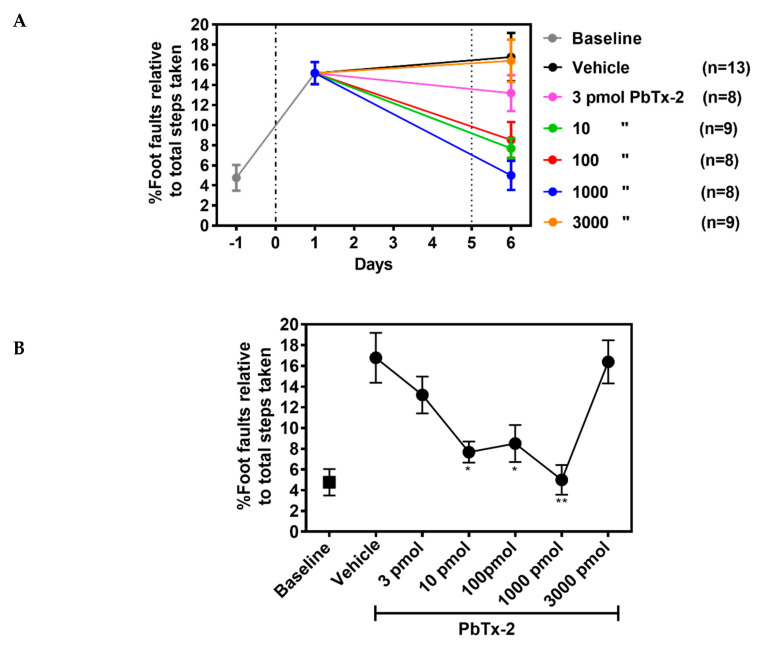
Effect of PbTx-2 treatment on foot fault task. (**A**) Animals were trained to walk on an elevated grid prior to photothrombotic stroke to obtain baseline values. On day 1 after inducing stroke, a significant increase in percentage of foot faults was observed. The 10, 100 and 1000 pmol doses of PbTx-2 produced significant improvement in percentage of foot faults as compared with the vehicle-treated animals. Alternatively, the 3 and 3000 pmol doses of PbTx-2 did not aid recovery. (**B**) Quantification of PbTx-2 dose–response effects on motor recovery 6 days post-stroke (one-way ANOVA followed by Dunnett’s post hoc test, 10 pmol PbTx-2 effect, * *p* = 0.023; 100 pmol PbTx-2 effect, * *p* = 0.027; 1000 pmol PbTx-2 effect, ** *p* = 0.001; 3 pmol PbTx-2 effect, *p* = 0.703; 3000 pmol PbTx-2 effect, *p* > 0.999). All data points are represented as mean ± SEM (n = 9 to 13 mice).

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
