# Peer review of "Epicortical Brevetoxin Treatment Promotes Neural Repair and Functional Recovery after Ischemic Stroke"

_marinedrugs, 2020, doi:10.3390/md18070374_

Round 1

Reviewer 1 Report

The authors of the article “Epicortical brevetoxin treatment promotes neuronal repair and functional recovery after ischemic stroke” describes how the use of brevetoxin leads to upreglation of NMDA receptor activity and eventually downstream Ca signaling pathways which lead to improved recovery of dendrites and synaptic density as well as motor recovery following a photothrombotic stroke model. Although interesting and well written, the article gives rise to a few questions. Please see below for queries and comments.

  1. The experimental design gives brevetoxin intracranially 5 days post the photothrombotic stroke was induced. Why wait so long to deliver a potential treatment? It is well known that PLA2 inhibitors need to be administered preferably 2 h following a stroke to have any effect. Not 5 days.
  2. It would be good to see another data point where the brevetoxin was administered much sooner after the photothrombotic stroke was induced, perhaps a day or so, to be able to compare the actual effect of brevetoxin.
  3. The authors give no explanation and no speculation for that matter why 3 and 3000 pmol of brevetoxin consistently failed to produce any improvement in any of the assays conducted? That 3 pmol may not be sufficient dose is one thing, but that 3000 pmol doesn’t do anything, and in Fig 7, 3000 pmol is worse than the vehicle.
  4. In figure legend for Fig 7 it is not explained for panel A what is going on. This explanation is following panel B. There is no explanation for panel B at what day the data was collected.
  5. In figure 7 it would have been beneficial to monitor the mice on some of the other days and the two data points look strange. Especially with 3000 pmol not producing any effect. Again, an additional dose of brevetoxin in a different group at day 1 or a few hours afer the photothrombotic stroke was induced would have improved any conclusion arising from this data.
  6. The same with Fig 6. The descriptions for Panel B and C are out of order and the authors don’t highlight at what time point panel C was collected. Again, no explanation for why 3000 pmol produced no result and the experiment would have been improved with additional data points following the photothrombotic stroke as well as an additional treatment point.
  7. There is very little focus on the cell biology behind the effect of the brevetoxin, or the route of administration. Although it is an interesting study, if the intracranial route of administration is very invasive.
  8. Do the authors think that there would have been a difference in infarct region if brevetoxin would have been administered earlier? Administering brevetoxin 5 days post photothrombotic stroke and then sacrificing the mice a day later seems like the infarct damage was done and the one day with bevetoxin wouldn’t be able to reverse the damage.

Minor points

  1. Please improve the use of abbreviations. PTS is used sometimes and only explained in a figure legend. The same thing with LF, RF, LH, RH is used in the text and only explained in the figure legend and then in the discussion. YFP is never described.
  2. Please add error bars to Fig 7
  3. On p 11, line 234 it says “with without”. It is unclear which one should be used?
  4. The article would be improved by a brief explanation in the results as to how the photothrombotic stroke was induced. This is not explained until the methods
  5. The figures could be drastically improved. They are not aligned and the x- and y-axis have different fonts/size within the same figure. Panels aren’t justified and some text looks incredibly big. They should be sized to fit into the paper without disrupting the flow of the paper. There is also text in the figures that are partially obscured by the different layers in the figures when it was assembled. Please address that to make sure no text is obscured.

Author Response

15 July 2020

Susie Deng

Assistant Editor

Marine Drugs

Dear Ms. Deng:

We thank the reviewers and editor for the thoughtful evaluation of our manuscript and have substantially modified the manuscript in accordance with their suggestions and comments. We were gratified to receive positive comments from both reviewers.

The following is a point by point-by-point response to all comments and suggestions of reviewer #1

Reviewer #1

  1. The selection of the time point of 5 days after a stroke was a critical component of our experimental design. This is due to neuronal excitability after a stroke exhibiting distinct phases during stroke progression and recovery. Therefore, enhancing cortical excitability too early after stroke may further increase neuronal death. The inflection point from the acute excitotoxic to chronic recovery phase occurs 3 days post-stroke in mice. This rational has been further amplified in the discussion section.

  1. As described above an earlier time point would likely be deleterious to recovery from stroke.

  1. An inverted-U model describes the relationship between NMDAR activity and neuronal survival and growth. This inverted-U NMDA concentration–response relationship has primarily, but not exclusively, been regressed to intracellular Ca2+ regulation. We have previously shown in cerebrocortical cultures that the effects of PbTx-2 on dendritic arborization and synaptogenesis displayed bidirectional concentration–response profiles (inverted-U). Moreover, PbTx-2 effects on calcium influx, dendritic arborization, and filopodia formation were dependent on NMDARs, underscoring the role of NMDAR signaling in the influence of PbTx-2 on neuronal structural plasticity (George et al., PNAS 109, 19840-45,2012). Therefore, the observed inverted-U does-response curves for epicortical treatment with PbTx-2 were predicted. These points have been amplified further in the manuscript.

  1. A more detailed description of the depicted data has been provided in the legend for Figure 7.

  1. We have addressed the importance of timing of the epicortical treatments above. Moreover, the lack of effect of the 3 and 3,000 pmol doses was again expected given the inverted-U concentration-response profiles previously reported for both PbTx-2 and NMDA in cerebrocortical cultures (George et al., PNAS 109, 19840-45,2012). This manuscript provides the first in vivo demonstration of an inverted-U does-response curve for PbTx-2.

  1. The clarity of the Figure 6 legend has been improved.

  1. We have added additional information regarding the molecular mechanisms of action of brevetoxin. The brevetoxin used, PbTx-2, is a well characterized sodium channel gating modifier that enhances sodium currents and influx. The use of epicortical or intracerebral cavity filling injections to treat strokes is a now well characterized strategy. We have added additional references to better indicate this. This method has been used clinically as well (see Kalladka et al., Lancet 388, 787-96, 2016).

  1. The issue of timing of epicortical treatment is addressed in 1 above.

Minor points:

  1. Abbreviations have been defined.
  2. Although we had not included for clarity, the errors bars have now been added to Figure 7.
  3. Corrected
  4. The photothrombotic stroke method is well recognized in the preclinical stroke literature and we feel the methods section is the proper place for description.
  5. We believe the formatting issues have been addressed.

Reviewer 2 Report

This manuscript contains interesting data to readers of the journal. In addition, the data were obtained and analyzed properly. However, just one comment on the results of the experiments. Authors should comment on why PbTx-2 showed no dose-dependencies in some results.

Author Response

15 July 2020

Susie Deng

Assistant Editor

Marine Drugs

Dear Ms. Deng:

We thank the reviewers and editor for the thoughtful evaluation of our manuscript and have substantially modified the manuscript in accordance with their suggestions and comments. We were gratified to receive positive comments from both reviewers.

The following is a point by point-by-point response to reviewer #2

Reviewer #2

1. PbTx-2 did display a dose dependency but the followed an inverted-U profile as previously shown in cerebrocortical neurons.

Round 2

Reviewer 1 Report

N/A

Author Response

Thank you so much for your valuable comments.